

# Feeding challenges in early infancy: the role of reflexes, muscle tone, and developmental milestones

Wiktoria Kowalska[1], Maria Tuczyńska[1], Jacek Kwiatkowski[2],
Oskar Komisarek[3], Ewa Mojs[4], Mirosław Andrusiewicz[5],
Tomasz Szczapa[6], Włodzimierz Samborski[1], Dorota Sikorska[1],
Ewa Baum[7] and Roksana Malak[1]

[1] Department and Clinic of Rheumatology, Rehabilitation and Internal Medicine, Poznan University of Medical Sciences, Poznań, Greater Poland, Poland
[2] Students Scientific Society of Maxillofacial Orthopaedics and Orthodontics, Poznan University of Medical Sciences, Poznań, Greater Poland, Poland
[3] Clinic of Otolaryngology, Laryngological Oncology and Maxillofacial Surgery, Dr. Jan Biziel Hospital, Bydgoszcz, Kuyavian-Pomeranian, Poland
[4] Department of Clinical Psychology, Poznan University of Medical Sciences, Poznań, Greater Poland, Poland
[5] Department of Cell Biology, Poznan University of Medical Sciences, Poznań, Greater Poland, Poland
[6] Neonatal Biophysical Monitoring and Cardiopulmonary Therapies Research Unit, II Department of Neonatology, Poznan University of Medical Sciences, Poznań, Greater Poland, Poland
[7] Department of Social Sciences and the Humanities, Poznan University of Medical Sciences, Poznań, Greater Poland, Poland

Corresponding authors
Maria Tuczyńska,
maria.tuczynska25@gmail.com
Roksana Malak, rmalak@ump.edu.pl

## ABSTRACT

**Background:** Problems with feeding are widespread in pre-term infants, but they also occur in full-term infants. Feeding skill is the first coordinated function a child acquires, providing him with nutrients and sensory stimulation needed for further proper development. The aim of the retrospective observational case-control study was to observe factors that may influence feeding problems in infants aged 0–3 months. The observed factors included the presence of oral reflexes, the differences in muscle tension, the advancement of motor development, the spontaneous movements, as well as the gestational age, and the pH value of the umbilical cord arterial.

**Methods:** The study involved 60 infants. The study and the control groups included 30 infants each. Feeding problems were the major inclusion criteria for the study group. The infants' reflexes were checked, the muscle tone was palpated, the presence of general movements was visually assessed using Prechtl's method, then the Sensitivity Assessment of the Stomatognathic Complex (SOWKUT) questionnaire and Albert Infant Motor Scale (AIMS) scale were evaluated. Through the perinatal interview with the patient's parent, information regarding the pH of the umbilical cord arterial value was obtained.

**Results:** The study involved 60 infants born between 24 and 41 gestational age (median and standard deviation: 35 ± 4.81). At the time of the study, their postconceptual age was 44 ± 7 weeks. The results showed that infants with problems regarding eating performance have their oral reflexes more often impaired, and their muscles more often show increased symmetrical tension, especially the frontal,
orbicularis oris, and masseter muscles. The influence of delayed motor development, sensory hypersensitivity, and early gestational age on the occurrence of feeding problems was observed. No correlation was observed between the umbilical cord arterial's pH values and the feeding issues.

**Conclusions:** Feeding problems are multifactorial, which implies that infants should be provided with quick intervention and necessary therapy. It will allow the babies to develop correctly and reduce the risk of future problems.

## INTRODUCTION

Research shows that 75% of premature infants and 23% of full-term infants have problems with eating (*Pineda, 2016*; *Malak et al., 2022b*). Oral feeding is the first activity an infant learns and is also necessary for discharging a child admitted to the Neonatal Intensive Care Unit (*Pineda et al., 2020a*). The conditions determining whether oral consumption of food is safe and effective are the risk of aspiration, the amount of food taken in time, as well as the integration of sucking, swallowing, and breathing processes affecting cardiorespiratory stability (*Lee et al., 2023*). Eating skills appear in infants around 33–34 weeks and depend on neurological development, influencing cognitive and motor responses through sensory experiences (*Pickler et al., 2005*).

The beginnings of oral feeding may be associated with problems resulting from immaturity or lack of reflexes supporting the eating function. These problems often disappear as the child matures, but they can lead to long-term issues and increase the risk of developmental disorders (*Pineda et al., 2020a*). According to *Kwon et al. (2020)*, the quality of eating skills in infancy is predictive of the development of these functions at the age of 4.

For the proper development of the eating function, it is crucial to integrate the frontal, temporal, medial, and lateral pterygoid muscles, masseter muscles, orbicularis oris, suprahyoid and infrahyoid muscles and muscles of the neck (*Basit, Eovaldi & Siccardi, 2023*).

We define a reflex as an automatic reaction of the nervous system to a sensory, proprioceptive, or auditory stimulus. From birth, reflexes accompany children in their everyday lives, allowing for spontaneous responses to previously unknown sensations and the surrounding world. They are critical for survival in a new environment (*Kondraciuk et al., 2014*).

Early examination of children for muscle tone, oral reflexes, or motor development is extremely important. Through these examinations, oral-motor exercises can be quickly introduced, which have an extremely beneficial effect on the feeding process, oral function, and oral-motor development. This therapy has no negative effects and can therefore be successfully used in neonatal wards (*Comuk Balci, Takci & Seren, 2023*).

The study aimed to observe the factors that may influence eating problems in infants aged 0–3 months and their impact on the proper development of this skill. The study's purposes also covered the investigation of the relationship between muscle tension, oral reflexes, and eating problems in infants. Moreover, motor development, oral reflexes association, gestational age, and oral reflexes were examined, and umbilical cord blood pH impact feeding difficulties.

Currently, very little researches focus on the underlying mechanisms that lead to feeding difficulties. Most studies concentrating on individual components, such as muscle tension or sensory hypersensitivity (*van Dijk, 2021*). *Jaafar et al. (2019)* identified the six most reliable questionnaires for assessing feeding in children with neurological conditions, underscoring the need for reliable diagnostic tools in this area. *Yang (2017)* highlighted the importance of a comprehensive approach, recommending thorough interviews, standardized questionnaires, and physical examinations, including anthropometric testing, to accurately assess feeding difficulties. Moreover, *Destriatania et al. (2025)* demonstrated the effectiveness of the top three questionnaires in evaluating feeding issues. Still, they emphasized that no single tool is sufficient to address the complexity of feeding disorders in children. Despite the valuable contributions of these studies, there remains a clear knowledge gap in the comprehensive assessment of feeding disorders, as they mainly focus on questionnaires. At the same time, additional diagnostic tools such as muscle palpation or facial reflex testing are also necessary to understand the full scope of these difficulties.

Given these gaps, our study aims to provide a holistic perspective, examining the relationship between oral reflexes, motor development, gestational age, and feeding difficulties. By addressing these underexplored connections, we aim to contribute to a more comprehensive understanding of the mechanisms underlying feeding problems, thus paving the way for future research that can improve early intervention strategies. Another aim was to evaluate the occurrence of selected oral and facial reflexes (rooting, sucking, phasic bite, snout, jaw jerk, and glabellar) in infants and assess their relationship with gestational age and motor development scores in order to help children in all age gain ability to be fed orally in the future.

## MATERIALS AND METHODS

This retrospective observational case-control study was conducted in the Greater Poland region in the Gynecology and Obstetrics Clinical Hospital of the Karol Marcinkowski Medical University in Poznań from January 2024 to May 2024 and involved 60 infants born between 24 and 41 gestational age. The flow diagram of the study is presented in Fig. 1. At the time of the study, their postconceptual age was 44 ± 7 weeks. The study and the control groups included 30 infants each. The major inclusion criteria for the study group were feeding problems. Moreover, to include them in the study, their parents needed to sign the written consent for participation and complete the questionnaires. The infants assigned to the control group did not present any feeding issues, had never been hospitalized, and their parents needed to sign the written consent for participation and complete the questionnaires. This group was made to compare the outcomes between both groups and find the factors influencing the proper feeding process. That is a wide range of
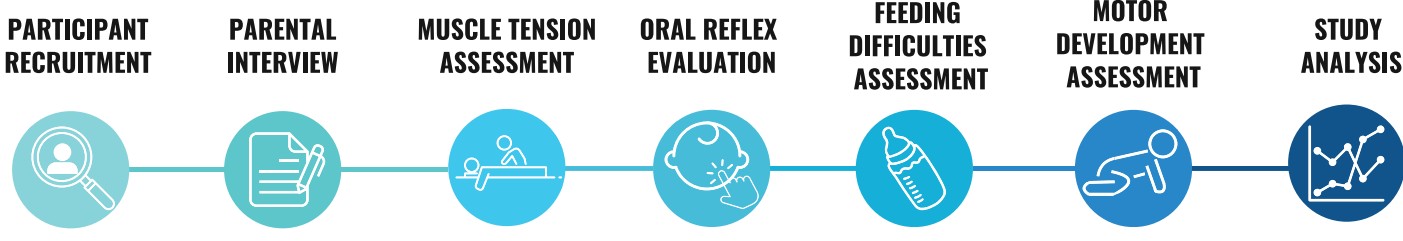

**PARTICIPANT RECRUITMENT** · **PARENTAL INTERVIEW** · **MUSCLE TENSION ASSESSMENT** · **ORAL REFLEX EVALUATION** · **FEEDING DIFFICULTIES ASSESSMENT** · **MOTOR DEVELOPMENT ASSESSMENT** · **STUDY ANALYSIS**

**Figure 1  Flow diagram.**                   

difficulties that includes: posseting, food aspiration while feeding, noneffective sucking (when suction is performed more frequently than swallows), problems with latch-on and suction on a bottle or a nipple, reluctance to feeding process itself, or if the infant takes too much milk at once. Exclusion criteria for both groups included babies with very low birth weight (<1,500 g), a saturation less than 88%, a heart rate less than 100 and higher than 205 beats per minute, active inflammation, septicemia, sepsis, tumors, encephalopathy, hypotension, and lethal congenital malformations.

The parents/legal guardians were first interviewed about the infant's birth history during the study. Afterward, we examined muscle tension related to feeding and assessed the presence of oral reflexes followed by feeding difficulties using the Sensitivity Assessment of the Stomatognathic Complex (SOWKUT) scale evaluation and motor development using the AIMS scale assessment. Finally, the overall movement patterns were checked. The entire examination lasted approximately an hour, which included describing the study to a parent or legal guardian, signing a consent, parental interview, and assessment of the infant.

Observations of the eating function began with obtaining perinatal information through an interview with the parent or legal guardian and the SOWKUT using two subscales. The first subscales assess oral and face hypersensitivity, while the second focuses on oral functions and anatomy. The infants' motor development was visually assessed using the Albert Infant Motor Scale (AIMS), and the development of the nervous system was evaluated by Prechtl's Assessment of Global Motor Patterns. The assessment of muscle tone was conducted through palpation of the examined muscle. We categorized them according to normal tone, asymmetrical hypertonia, and symmetrical hypertonia. The presence of oral reflexes was also evaluated through palpation to determine whether the reflex was present.

The SOWKUT commercially available tool is a standardized diagnostic tool supporting the work of therapists and a speech therapist/neurologist to enable early diagnosis and intervention regarding sensory problems affecting the child's face and masticatory organs, as well as issues with the child's food intake. Children from birth to 3 years of age can be assessed with this scale (*Wiśniewska & Kaczyńska, 2022*).

The Alberta Infant Motor Scale is a validated scale that allows the gross motor skills of developing infants to be assessed through observation. It assesses the achievement of milestones in the infant's motor development from birth to time when they gain the ability

to walk independently around 18 months of age (*do Fuentefria, Silveira & Procianoy, 2017*; *Piper & Darrah, 2021*). In our study, we used the percentile ranks for children in the Polish population, which are from the standardized Polish version of the AIMS scale (*Eliks et al., 2023*).

The AIMS scale is tested by observing the movement patterns presented by the child in four positions. When the motor pattern is present, the infant receives points corresponding to the advancement of the given motor pattern, while when the pattern is absent, the child gets 0 points (*Malak et al., 2022a*). After that, we compare the result of the assessment to their age and observe the outcome on the percentile scale that enables us to determine whether the child's motor development is delayed or not.

Assessment of global patterns, according to Prechtl, is also made by observing the infant and its spontaneous movements, paying attention to their quality, fluency, frequency, speed, alternation, and multidirectionality (*Malak et al., 2022a*).

Prechtl's Assessment of Global Movement Patterns is a tool that involves observing infants' spontaneous movements for approximately 3 to 5 min. It allows for predicting the neurological development of premature babies, full-term newborns, or children at risk (*do Fuentefria, Silveira & Procianoy, 2017*; *Piper & Darrah, 2021*).

The examination was observational, non-invasive, and painless. It does not differ from a standard physiotherapy procedure. All parents were informed before they gave written consent for their kid's participation in the study. The study was conducted in accordance with the Declaration of Helsinki, and the protocol was approved by the Poznan University of Medical Sciences Institutional Review Board (no. 26/24, date of approval January 10, 2024).

The study aimed to achieve comparable groups, thus enhancing the validity and reliability of the results. To observe differences between the study and control group, the approach we used contributed to minimizing confounding variables, allowing for a clearer interpretation of the impact of feeding problems on the clinical outcomes assessed in the study.

## Statistical analyses

Calculations were performed using Statistica version 13 for Windows OS (TIBCO Software, Toulouse, OK, USA) and PQStat version 1.8.6 (PQStat Software, Poznan, Poland). The following tests were used in the study: Chi-square ($\chi^2$) test of independence tests according to Cochran's rules with odds ratio/risk ratio (OR/RR) and 95% confidence interval (95% CI) estimation and also tetrachoric correlation ($r_t$), and ANOVA Kruskal Wallis test with the Dunn Benjamini-Hochberg *post hoc* test. The sample size for this study was determined based on the comparison of two proportions. The expected proportion of infants exhibiting feeding problems in the study group was estimated to be 33% (0.33), while the expected proportion in the control group was estimated to be 67% (0.67). A significance level of 0.05 was chosen for the study. The desired statistical power was set at 0.80. As a result, the calculated sample size indicated that approximately 30 infants were required in each group to detect a statistically significant difference between the study and control groups at the specified alpha level and power. Based on these calculations, the study

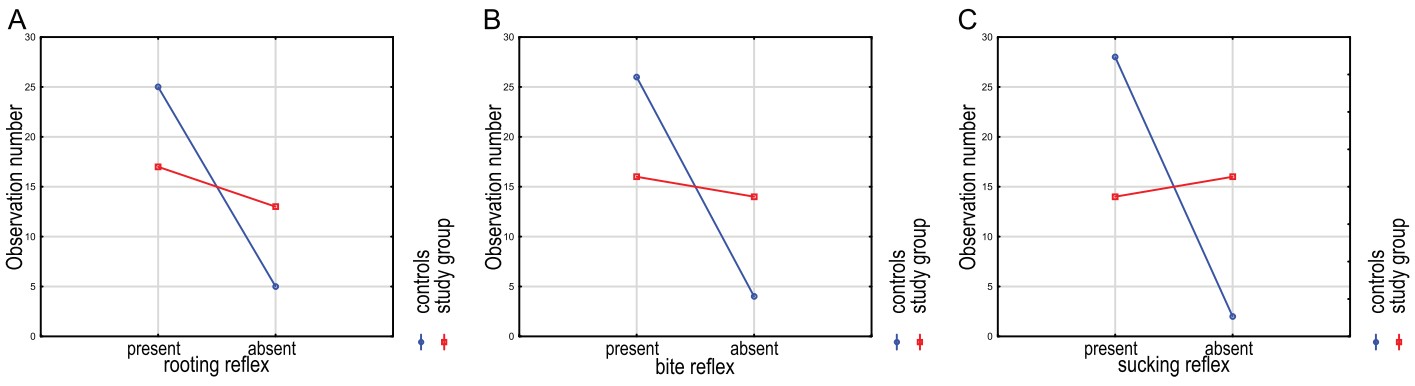

**Figure 2** The presence or absence of the rooting (A), bite (B), and sucking (C) reflexes in examined children.

included 60 infants, with 30 allocated to each group, ensuring adequate power to detect the hypothesized effects. The level of statistical significance was set at $p < 0.05$.

## RESULTS

A palpation examination was performed to check the presence or absence of the rooting reflex. The occurrence of the rooting reflex was significantly different between the control and study groups ($\chi^2$ ($df = 1$, $N = 60$) = 5.22, $p = 0.0224$; OR 3.82, 95% CI [1.15–12.71]; $r_t = 0.47$). In the study group, the rooting reflex could be observed less likely (17 infants, and in the control group in 25; Fig. 2). The occurrence of the rooting reflex in the study group was 40% and in the control group 60%, and the absence was 72% and 28%, respectively. The chance of rooting reflexes presence increased 3.82-fold in the control group. Differences in reflexes observed between study groups are presented in the Table 1.

The phasic bite reflex appeared in 16 children in the study group and in 26 children in the control group. The results of the chi-square test indicate a statistically significant difference between the study and control groups and the occurrence of the phasic bite reflex ($\chi^2$ ($df = 1$, $N = 60$) = 7.94, $p = 0.0099$; OR 5.69, 95% CI [1.59–20.33]; $r_t = 0.58$). The reflex occurred in 42 tested infants, including 62% of children in the control group and 38% in the study group (Fig. 2). The chance of a phasic bite reflex presence increased 5.69-fold in the control group.

The presence of the sucking reflex, compared to both groups, was recorded in 33% of infants from the study group and 67% of infants from the control group. There was a statistically significant difference between the study and control groups in the occurrence of the sucking reflex ($\chi^2$ ($df = 1$, $N = 60$) = 15.56, $p < 0.0001$; OR 16.00, 95% CI [3.22–79.56]; $r_t = 0.77$). Comparing both groups regarding the occurrence of this reflex, it appeared in 28 children from the control group and 14 infants from the study group (Fig. 2). The chance of the presence of a sucking reflex increased 16-fold in the control group.

The remaining analyses regarding snout, jaw jerk, and glabellar reflexes showed insignificant differences in the case-control study ($p > 0.05$).

**Table 1 Differences in reflexes observed between study groups.**

| Reflexes | p-value |
| --- | --- |
| **Differences between the study and control groups in assessed reflexes** | |
| Rooting reflex | 0.0224[a] |
| Sucking reflex | <0.0001[a] |
| Phasic bite reflex | 0.0099[a] |
| Snout reflex | N/S[a] |
| Jaw jerk reflex | N/S[a] |
| Glabellar reflex | N/S[a] |
| **Differences between low and high AIMS score in assessed reflexes** | |
| Rooting reflex | 0.0356[b] |
| Sucking reflex | N/S[b] |
| Phasic bite reflex | N/S[b] |
| Snout reflex | N/S[b] |
| Jaw jerk reflex | N/S[b] |
| Glabellar reflex | N/S[b] |
| **Differences between low and high SOWKUT score in assessed reflexes** | |
| Rooting reflex | 0.0004[b] |
| Sucking reflex | 0.0001[b] |
| Phasic bite reflex | 0.0002[b] |
| Snout reflex | 0.0003[b] |
| Jaw jerk reflex | 0.0009[b] |
| Glabellar reflex | 0.0002[b] |
| **Differences in assessed reflexes according to gestational age** | |
| Rooting reflex | 0.0034[b] |
| Sucking reflex | 0.0036[a] |
| Phasic bite reflex | 0.0026[a] |
| Snout reflex | N/S[a] |
| Jaw jerk reflex | N/S[a] |
| Glabellar reflex | N/S[a] |

Notes:
[a] Chi$^2$ test.
[b] Kruskal-Wallis ANOVA test.
N/S, not significant ($p > 0.05$).

The percentile result of the Albert Infant Motor Scale test significantly impacts the rooting reflex in the control and study groups ($H$ ($df = 3$, $N = 60$) = 8.57; $p = 0.0356$). The differences were only observed between the controls with present and absent rooting reflexes ($p = 0.0447$). As the percentile value decreases, the occurrence of the rooting reflex in the tested infants decreases (Fig. 3).

The results of the Kruskal Wallis analysis of variance (ANOVA) test did not show a significant difference in the percentile value of the AIMS scale and the occurrence of the sucking, phasic bite, snout and jaw jerk, and glabellar reflexes in the case-control study ($p > 0.05$).

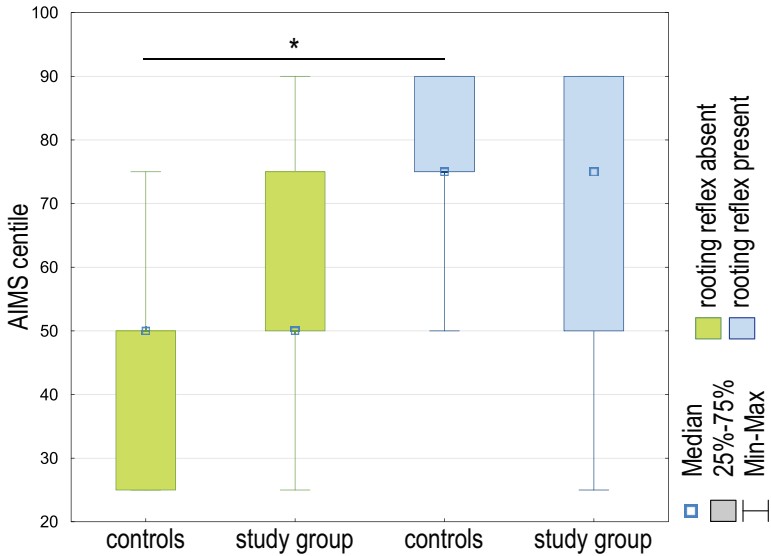

**Figure 3 The AIMS centile differences between the control group and the study group are regarding the presence or absence of the rooting reflex in examined children.** *$p < 0.05$.

Significant differences were established in the SOWKUT scale's percentile score and the rooting reflex's occurrence in the case-control study ($H$ ($df$ = 3, $N$ = 60) = 18.19; $p$ = 0.0004). The observed differences were estimated between controls and study groups without rooting reflex ($p$ = 0.02032) as well as controls with present rooting reflex and both study groups—with rooting reflex ($p$ = 0.0167) and absent ($p$ = 0.0009). The highest median SOWKUT values were in cases lacking rooting reflex, and the lowest in controls presented the reflex (Fig. 4).

It was observed that most children with normal rooting reflexes had higher percentile scores. Similar results could be observed in the case of the phasic bite reflex ($H$ ($df$ = 3, $N$ = 60) = 20.05; $p$ = 0.0002), snout reflex ($H$ ($df$ = 3, $N$ = 60) = 18.88; $p$ = 0.0003), jaw jerk reflex ($H$ ($df$ = 3, $N$ = 60) = 16.51; $p$ = 0.0009), glabellar reflex ($H$ ($df$ = 2, $N$ = 60) = 16.61; $p$ = 0.0002), and sucking reflex ($H$ ($df$ = 3, $N$ = 60) = 21.61; $p$ = 0.0001). Significant differences were found between the SOWKUT scale's percentile score and reflexes' presence in all cases. In the case of the bite reflex, the differences were observed between the study group without the bite reflex and both control groups—without the reflex ($p$ = 0.0189) and presenting the bite reflex ($p$ = 0.0002; Fig. 4). Regarding the snout reflex, it was similar, the differences were observed between the study group lacking bite reflex and both control groups—without the reflex ($p$ = 0.0253) and presenting the bite reflex ($p$ = 0.0013) and additionally, study group and controls presenting the snout reflex ($p$ = 0.0042; Fig. 4). Considering the jaw jerk reflex, we observed differences between the study group presenting the reflex and both controls—with absent jaw jerk reflex ($p$ = 0.0385) and positive for this reflex ($p$ = 0.0011; Fig. 4). There was no case of glabellar reflex negative in the control group. Instate the controls presenting the glabellar reflex; the SOWKUT level differed with both control ($p$ = 0.0456) and cases ($p$ = 0.0004) presenting

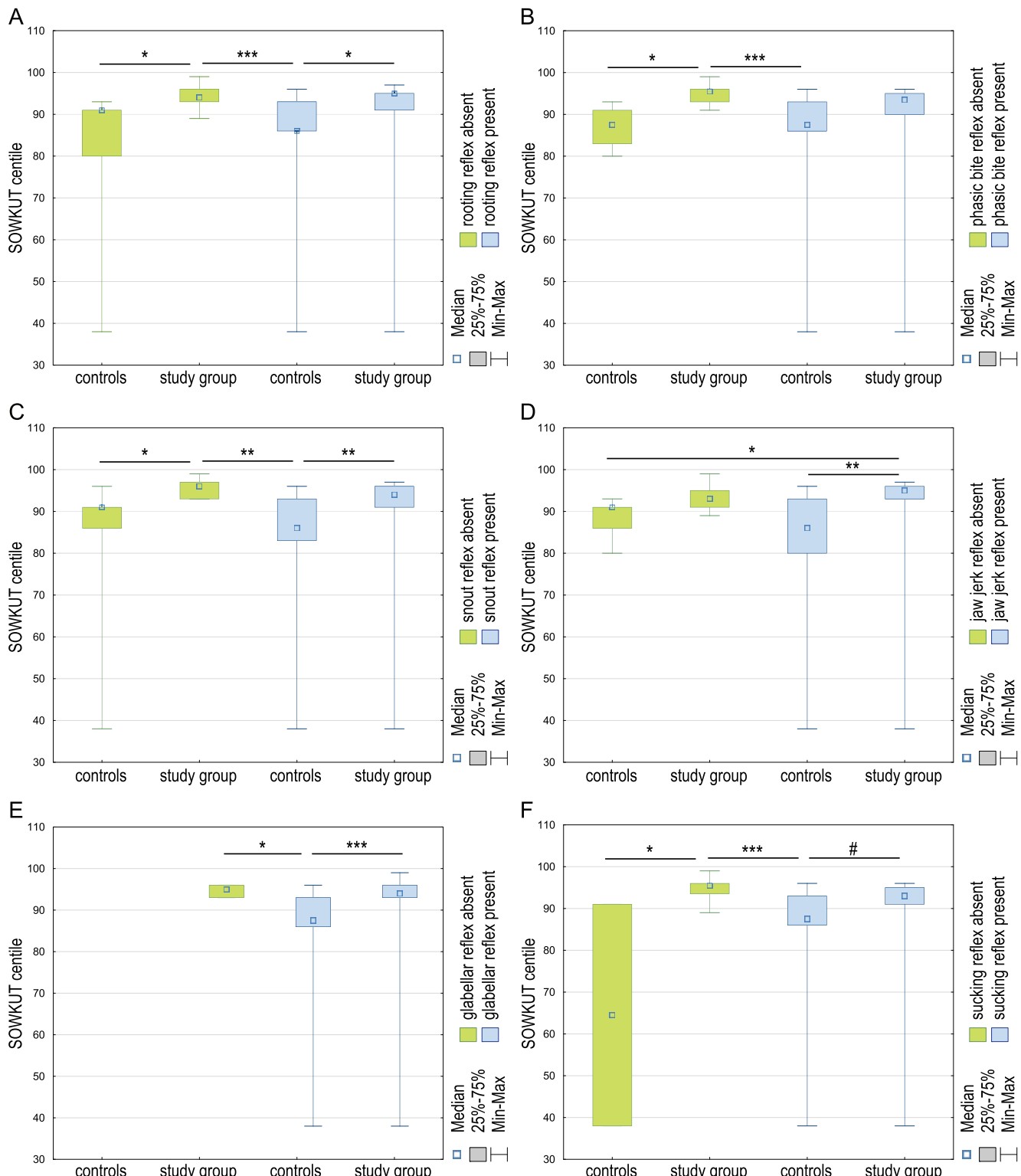

**Figure 4** The SOWKUT centile differences between controls and study group regarding the presence or absence of the rooting (A), phasic bite (B), snout (C), jaw jerk (D), glabellar (E), and sucking (F) reflexes in examined children. $^{\#}p < 0.1$, $^{*}p < 0.05$; $^{**}p < 0.01$; $^{***}p < 0.001$.

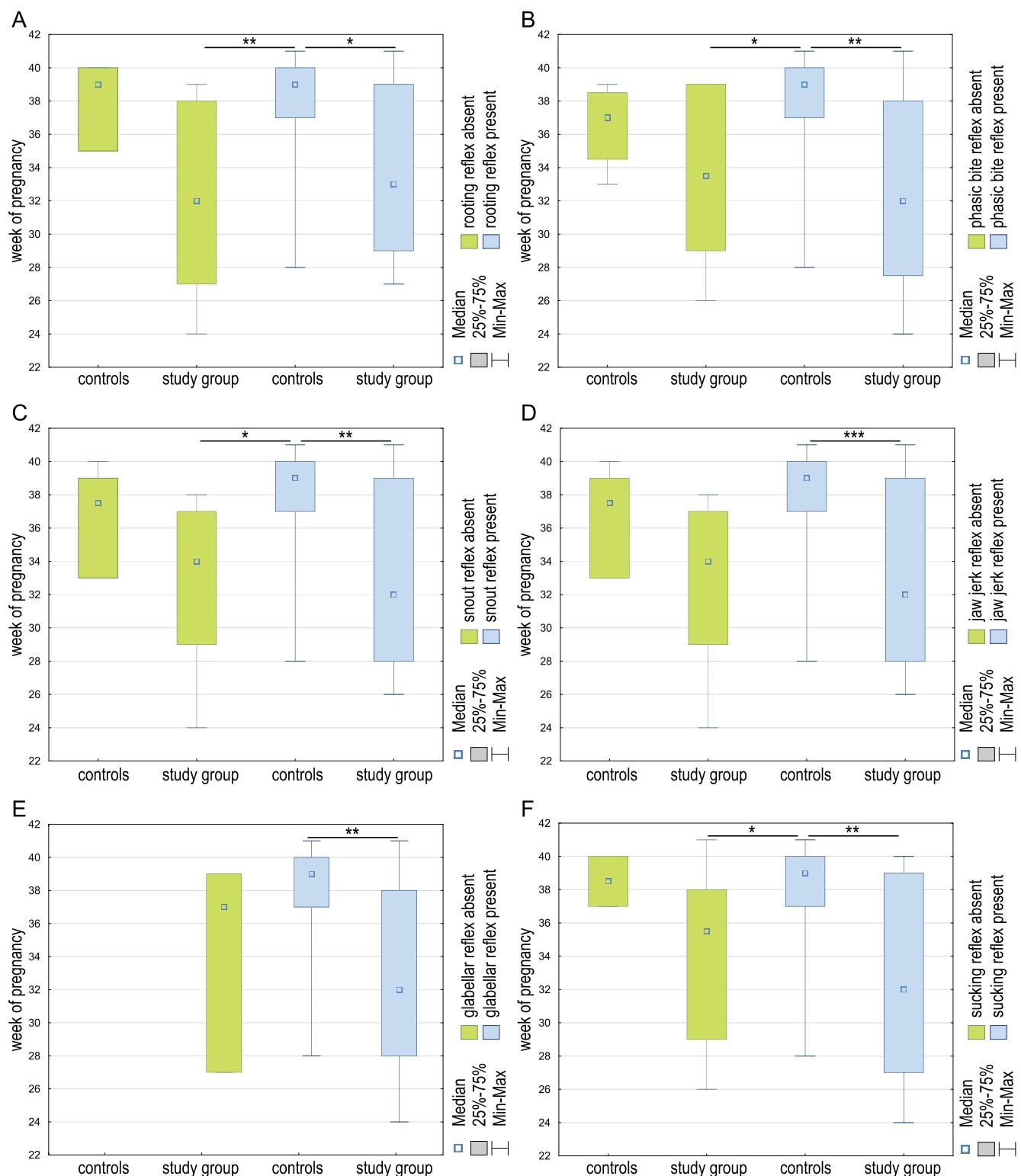

**Figure 5** The week of pregnancy differences between the controls and study group regarding the presence or absence of the rooting (A), phasic bite (B), snout (C), jaw jerk (D), glabellar (E), and sucking (F) reflexes in examined children. *$p < 0.05$; **$p < 0.01$; ***$p < 0.001$.

this reflex (Fig. 4). In the case of the sucking reflex, the differences were observed between both control groups—without the reflex ($p = 0.0408$) and presenting the sucking reflex ($p < 0.0001$). Additionally, there was a slight (but not significant; $p = 0.0555$) difference in SOWKUT centile level between both control groups (Fig. 4).

Significant differences between the week of pregnancy in which the infant was born and the occurrence of the rooting reflex ($H$ ($df = 3$, $N = 60$) = 13.64; $p = 0.0034$) were observed. The later the baby was born, the more often a normal reflex could be observed. Significant differences were observed between controls positive for the rooting reflex and both study groups—presenting the reflex ($p = 0.0344$) and without the reflex ($p = 0.0083$; Fig. 5). Similar differences were observed for the phasic bite reflex ($H$ ($df = 3$, $N = 60$) = 14.21; $p = 0.0026$). It has been shown that abnormal or absent reflexes may be seen more often in babies born earlier. Significant differences were observed between controls positive for the phasic bite and both study groups—presenting the reflex ($p = 0.0049$) and without the reflex ($p = 0.0134$; Fig. 5). Also, differences in rooting reflex ($H$ ($df = 3$, $N = 60$) = 14.12; $p = 0.0027$) were shown between controls positive for the rooting reflex and both study groups—presenting the reflex ($p = 0.0056$) and without the reflex ($p = 0.0178$; Fig. 5). The difference in jaw jerk reflex ($H$ ($df = 3$, $N = 60$) = 8.70; $p = 0.035$) was observed between controls and study cases ($p = 0.0005$) presenting the jaw jerk reflex. Pregnancy week differed between children in the case-control study when comparing the presence or absence of the glabellar reflex ($H$ ($df = 2$, $N = 60$) = 13.17; $p = 0.0015$), and the difference was established between presenting the reflex controls and study participants ($p = 0.0014$; Fig. 5). The differences were also observed for the sucking reflex ($H$ ($df = 3$, $N = 60$) = 13.52; $p = 0.0036$) and the phasic bite reflex. The controls positive for the sucking reflex differed with both study groups—sucking reflex negative ($p = 0.0322$) and positive ($p = 0.0070$; Fig. 5). It has been shown that abnormal or absent reflexes may be seen more often in babies born earlier.

## DISCUSSION

Understanding and addressing feeding difficulties in infants, particularly regarding preterm populations, requires an in-depth analysis of underlying mechanisms. Our study highlights the relationship between oral reflexes and motor development, emphasizing that weaker or absent reflexes are associated with feeding problems. The findings contribute to the growing body of evidence indicating that feeding difficulties in infancy are often multifactorial, involving neuromuscular, sensory, and gestational factors. Supporting children with eating problems should start with finding and correctly understanding their causes, and we should remember that they are often complex. Many aspects should be considered when assessing the feeding process in preterm infants. Studies report that the issues may be caused by increased or decreased muscle tension in the masticatory system, sensory hypersensitivity in this area, pathology of oral reflexes, delayed motor development, or related to an earlier birth week (*Hawdon et al., 2000*; *Bauer et al., 2008*; *Porges, 2023*). The ability to eat is one of the first coordinated activities of an infant to ensure its further development by providing nutrients (*Seward & Serdula, 1984*; *Dewey, 2001*; *Park et al., 2014*).

The problem with developing these functions will affect their further growth (*Crapnell et al., 2015*). Possessing these skills is also an important signal related to the maturity of the nervous system; their absence may indicate severe dysfunctions (*Mizuno & Ueda, 2005*; *Malak et al., 2022b*).

The lack of coordination of the processes supporting eating may result from poorer control over postural muscles, the muscles of the masticatory system, or hypotonia (*Taylor & Emmett, 2019*). The work of the masseter muscles is essential in the sucking process, allowing the infant to obtain food by sucking the nipple or bottle teat (*Inoue, Sakashita & Kamegai, 1995*; *França et al., 2014*). The increased activity of the masseter muscles may negatively affect oral functions, including the sucking reflex. As the masseter muscles mature, they adapt to the changing structures of food given to the child (*Abadie & Couly, 2013*). Therefore, their proper functioning is critical in developing the eating function (*Inoue, Sakashita & Kamegai, 1995*; *França et al., 2014*).

Oral reflexes are abnormal in children with eating problems (*Bauer et al., 2008*; *Black, 2012*). A significant relationship was observed between these reflexes' weakening, arrhythmic or absence, and difficulties starting the feeding process. It leads to abnormalities related to the coordination of the sucking-swallowing-breathing process, causing rapid fatigability in infants (*Wahyuni et al., 2022*).

Infants with an absent rooting reflex were twice as likely to have feeding problems as infants in whom the rooting reflex was present (*Wahyuni et al., 2022*). The presence of the feeding reflex is considered a sign of the baby's readiness to begin the independent feeding process. The weakened rooting reflex correlates with the tongue function, consequently affecting the sucking reflex. Sucking becomes arrhythmic and disorganized, causing reduced effectiveness of this function (*Thoyre, Shaker & Pridham, 2005*). The integrated vagal system regulates heart rate, facilitating the sucking-swallowing-breathing process (*Porges, 2023*).

Problems with the sucking-swallowing process may be an alarm signal about changes in the brain stem. Persistent feeding difficulties may also indicate congenital neuromuscular diseases and complicate the course of various progressive or non-progressive neurological diseases (*Abadie & Couly, 2013*).

Infants' motor development affects their oral reflexes. Eating difficulties were observed more often in infants with delayed motor development. The lack of coordination of processes supporting eating may result from poorer control over postural muscles, the muscles of the masticatory system, or hypotonia (*Erol et al., 2023*). In infants with lower scores on the Alberta Infant Motor Scale, motor and anti-gravity skills difficulties prevented the stomatognathic system from functioning correctly. Improvement in motor functions was observed in children who received feeding support therapy (*Erol et al., 2023*). The disorganized sucking reflex in children at 37 weeks of corrected age was accompanied by a delay in the motor skills development measured at six months of age (*Tsai, Chen & Lin, 2010*).

The relationship between the occurrence of eating difficulties and sensory hypersensitivity was observed. Three areas were identified in which the problem of processing sensory stimuli occurred the most. These include tactile, vestibular, and oral

hypersensitivity (*Yi et al., 2015*). Similar results were also found in a study by *Farrow & Coulthard (2012)*. These researchers draw attention to the problem induced by sensory hypersensitivity, which affects feeding problems associated with selective eating at a later age, as well as a significant correlation with the co-occurrence of anxiety in children. Infants fed enterally for about three weeks may develop issues with sensory hypersensitivity and delays in swallowing skills, which increases the risk of food aspiration. Hypersensitivity in the face area induces defensive behaviors related to stress in the child, causing crying, intensifying the vomiting reflex, or turning the head at the sight of a bottle or nipple (*Dodrill et al., 2004*).

In children with a lower gestational age, impaired oral reflexes may be observed and negatively affect eating skills. Babies born prematurely have a lower degree of arousal, which means that it is more difficult for them to maintain their feeding, which affects the safety and effectiveness of this process (*Pineda et al., 2020b*). In the research of *Hawdon et al. (2000)*, a relationship between a lower gestational age and a reduced presentation of the eating function was found. The impact of these problems on their subsequent functioning at the age of 6 months was also noted. The vomiting reflex was six times more common in children with difficulties in sucking-swallowing-breathing. Choking problems occurred three times more often during feeding in this group.

*Wahyuni et al. (2022)* confirmed that lower gestational age increases the likelihood of oral reflex problems affecting the correct eating function presentation.

In infants born less than 33 weeks of age, the sucking reflex was found to be chaotic and with low amplitude. During maturation, the sucking reflex stabilizes, and its frequency increase, making it more organized. It allows them to develop the correct sucking-breathing-swallowing pattern (*Gewolb et al., 2001*).

Our study provides valuable insights into the potential causes of infant feeding difficulties. Using the AIMS and SOWKUT scales, muscle palpation, reflex examinations, and perinatal history (including umbilical cord blood pH and gestational age), we identified key factors contributing to feeding problems. Infants with lower AIMS scores and increased facial muscle tension were more likely to experience feeding difficulties. Similarly, higher SOWKUT scores—indicating sensory hypersensitivity—were associated with increased feeding challenges.

Although our findings on umbilical cord blood pH were inconclusive, previous research by *Malak et al. (2021)* suggests that this factor may influence sucking reflex development. Future studies should further explore the role of perinatal factors in early feeding dysfunction.

Unlike previous studies focusing primarily on the neuromuscular or sensory aspects of feeding difficulties, our research integrates motor and gestational factors, bridging a gap in the literature. By demonstrating the statistical relationship between reflex presence, motor development, and gestational age, we reinforce the importance of assessing feeding difficulties within a broader developmental context.

Given the complexity of feeding disorders, an interdisciplinary approach—including pediatric neurology, occupational therapy, and speech therapy—may be the most effective strategy for intervention. Future research should investigate the long-term impact of

feeding difficulties on cognitive and behavioral outcomes and the effectiveness of early therapeutic interventions.

## CONCLUSIONS

Our study confirms that oral reflexes are essential for proper and effective feeding. The significant differences between the study and control groups emphasize the need for early assessments of oral reflexes and motor function, particularly in preterm infants. Targeted interventions focusing on motor skill development may help mitigate feeding difficulties in at-risk populations. Preterm infants present significantly lower occurrence of the rooting, sucking, and phasic bite reflexes compared to the control group. The presence of these reflexes strongly correlates with higher motor development scores (AIMS, SOWKUT) and later gestational age. Abnormal or absent reflexes occur more often in preterm infants. Future research should explore intervention strategies that support oral motor function in preterm infants to improve feeding outcomes.

**Limitations of the study:** Primary limitation of this study is the small sample size, which resulted in a moderate evidence level. Future research with a larger and more diverse sample is needed to confirm these preliminary observations. The study was limited by the lack of long-term follow-up and potential variability in reflex assessment due to the subjective nature of clinical palpation.

## ACKNOWLEDGEMENTS

This study was undertaken with the contribution of midwives and neonatologists from the neonatology clinic, who were involved in the preparation of each neonate for the assessment. Special thanks to the midwives and neonatologists at the Gynecology and Obstetrics Hospital in Poznań. We acknowledge the use of Grammarly (version 1.2.135.1595) to check for spelling and grammatical errors and for enhancing text readability. ChatGPT was used for English correction and rephrasing in some sections of the manuscript. We reviewed the changes critically and, based on this, revised the writing.

### Funding

The authors received no funding for this work.

### Competing Interests

The authors declare that they have no competing interests.

### Author Contributions

- Wiktoria Kowalska conceived and designed the experiments, performed the experiments, analyzed the data, authored or reviewed drafts of the article, and approved the final draft.
- Maria Tuczyńska conceived and designed the experiments, performed the experiments, authored or reviewed drafts of the article, and approved the final draft.

- Jacek Kwiatkowski performed the experiments, authored or reviewed drafts of the article, and approved the final draft.
- Oskar Komisarek conceived and designed the experiments, authored or reviewed drafts of the article, and approved the final draft.
- Ewa Mojs performed the experiments, authored or reviewed drafts of the article, and approved the final draft.
- Mirosław Andrusiewicz analyzed the data, prepared figures and/or tables, authored or reviewed drafts of the article, and approved the final draft.
- Tomasz Szczapa conceived and designed the experiments, authored or reviewed drafts of the article, and approved the final draft.
- Włodzimierz Samborski conceived and designed the experiments, authored or reviewed drafts of the article, and approved the final draft.
- Dorota Sikorska conceived and designed the experiments, authored or reviewed drafts of the article, and approved the final draft.
- Ewa Baum conceived and designed the experiments, authored or reviewed drafts of the article, and approved the final draft.
- Roksana Malak conceived and designed the experiments, performed the experiments, authored or reviewed drafts of the article, and approved the final draft.

## Human Ethics

The following information was supplied relating to ethical approvals (*i.e.*, approving body and any reference numbers):

Poznan University of Medical Sciences Institutional Review Board (no. 26/24, date of approval January 10, 2024).

## Ethics

The following information was supplied relating to ethical approvals (*i.e.*, approving body and any reference numbers):

Poznan University of Medical Sciences Institutional Review Board (no. 26/24, date of approval January 10, 2024).

## Data Availability

The raw measurements are available in the Supplemental File.

## Supplemental Information

Supplemental information for this article can be found online at http://dx.doi.org/10.7717/peerj.19777#supplemental-information.

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
