# Peer review of "Feeding challenges in early infancy: the role of reflexes, muscle tone, and developmental milestones"

_PeerJ, doi:10.7717/peerj.19777_

## Round 0.1 · original submission · Major Revisions

The manuscript addresses an important topic, but significant revisions are needed to meet publication standards. The language is clear, and the manuscript is well-structured. Reviewers highlighted several methodological issues, including the lack of a sample size calculation, insufficient details on randomization, and unclear descriptions of the control group. The limitations, such as the small sample size and low evidence level, should be explicitly acknowledged. Overall, while the study has clinical relevance, substantial revisions are required to enhance its validity and impact.

·

Basic reporting

Thank you authors for investigating this important problem seen in neonatal period, however, I have some concerns about the study.
Here are some examples:

Abstract:
- Does standard deviation have no decimals?
- They should give statistical results and the p values in results section.
Materials and Methods
- Please explain the feeding problems for the inclusion criteria
- Explain about the randomization system of the study
- Explain more about the outcome measures and their scoring
- Explain about the sample size calculation of the study
- Please add flow diagram of the study

Experimental design

In the whole, the level of evidence of the study is not enough for publication.

Validity of the findings

The evidence quality of the study is low in terms of nutritional parameters and randomization. The limitations and uncertainties of the study have negatively affected the quality of the evidence.

Reviewer 2 ·

Basic reporting

Title
The current title of the article, "Oral-Motor Dysfunctions and Neonatal Factors Associated with Feeding Problems in Infants Aged 0-3 Months," does not fully reflect the specific focus and methodology of the study. Based on the abstract, it is evident that the research primarily examines preterm infants rather than all neonates. Furthermore, the case-control design of the study is a critical methodological aspect that should be explicitly mentioned in the title to provide clarity and context for potential readers. Given these considerations, a revised title is proposed: "Oral-Motor Dysfunctions and Neonatal Factors Associated with Feeding Problems in Preterm Infants: A Case-Control Study."

Introduction
From lines 83 to 97 of the introduction, the topics discussed should be moved to the Materials and Methods section, as they describe the study methodology. The same concepts can be summarized with a single sentence if the authors wish to retain the theme in the introduction, stating that various scales and methods exist to measure alterations and performance, etc.

M&M
In the methodology section, it is recommended to add specific subsections: Sample Size Calculation, providing details on how the sample size was determined; Study Group, Control Group, and Inclusion and Exclusion Criteria, clearly defining these aspects. This will make the text more accessible and easier for readers to follow. Additionally, include subsections for Study Duration and Times of Follow-Up. These subsections can be organized with individual subtitles and subparagraphs for clarity.

Discussion
The discussion is too lengthy and needs to be shortened by at least half.

Experimental design

The methodology section needs to be improved as outlined in the revisions.

Validity of the findings

As specified in the comments, the sample size calculation must be detailed.

Additional comments

No additional comments.

Reviewer 3 ·

Basic reporting

- The English language is clear and easy to understand. An adequate number of references has been provided. The article's structure, figures, and tables are appropriately structured.

Experimental design

- Please elaborate further on the role of the control group in the study methods.

Validity of the findings

- The discussion section seems to merely compile previous studies. Please elaborate in detail on the significance of your study findings and how the results address existing knowledge gaps.
- Revise the conclusion section to clearly state your study results and recommendations for intervention targeting the population. The authors should consider omitting citations in this section.

Additional comments

- Please outline the existing knowledge gaps on this topic in the Introduction section and emphasize the necessity for further studies.

---

## Round 0.2 · Major Revisions

Dear authors,

Thank you for submitting the revised version of your manuscript, "Feeding challenges in early infancy: The role of reflexes, muscle tone, and developmental milestones", and for your thoughtful responses to the reviewers’ comments.

I appreciate the effort you have put into improving the clarity and structure of the manuscript. However, one reviewer and myself noted a critical issue related to the description of the study design that needs to be addressed before the manuscript can proceed further in the review process.

The manuscript repeatedly refers to a randomized design, suggesting that participants were randomly allocated to groups. However, based on your description, the study appears to be a retrospective observational case-control design, in which infants were grouped based on the presence or absence of feeding problems. There is no indication that random assignment was used.
This distinction is important, as the term randomization carries specific methodological implications related to bias control and inference strength. Mislabeling an observational design as randomized could be misleading to readers and does not align with expected reporting standards.

I kindly ask that you revise the manuscript to: 1. Accurately describe the study design as observational (case-control), removing all references to randomization. 2. Clearly explain the criteria for group selection and any matching procedures used. 3. Ensure that conclusions and interpretations are aligned with the limitations of an observational design.

Once these revisions are made, I will be happy to reconsider the manuscript for further review.

Additionally, and just as an open suggestion, I would be happy to see at least one table summarizing the results, which could help improve the understanding of your results.

I look forward to receiving your updated version.

·

Basic reporting

-I am not sure if the results added some new information for the literature so i reject to accept for publication of the study.

Experimental design

-Although the study is an observational study, they mentioned as it was an randomized controlled clinical trial
-Also they didn't mention enough about the randomization sytem.

Validity of the findings

The findings are ok.

---

## Round 0.3 · Minor Revisions

Thank you for your resubmission with corrections. Upon reviewing Figure 1 in your manuscript, I noticed that it still includes the design with randomization that requires correction. I kindly request that you revise this figure to accurately reflect the appropriate design.

While waiting for one reviewer to answer to your last version, please submit the corrected figure at your earliest convenience. Once I receive the updated version, we will proceed with the next steps in the review process.

Kind regards

·

Basic reporting

It is proper and acceptable.

Experimental design

It is proper and acceptable.

Validity of the findings

It is proper and acceptable.

---

## Round 0.4 · accepted · Accept

Dear Authors,
I am pleased to accept your article!
As a suggestion, however, I recommend removing Figure 1, as I still find it misleading and not reflective of your methodology.
Kind regards,

Reviewer 3 ·

Basic reporting

- The English language is clear and easy to understand. An adequate number of references has been provided. The article's structure, figures, and tables are appropriately structured.

Experimental design

The authors addressed my concerns

Validity of the findings

Discussion section was revised to incorporate my suggestions

Additional comments

None